# Warrior Gods and Otherworldly Lands: Daoist Icons and Practices in Late Chosŏn Korea

## Maya Stiller

College of Liberal Arts & Sciences, The Kress Foundation Department of Art History, University of Kansas, Lawrence, KS 66045, USA; mstiller@ku.edu

**Abstract:** This article brings Chosŏn dynasty (1392–1910) Korea into the discussion about the various roles of Daoism in East Asian cultures in which it has, unfortunately, all too often been absent. Based primarily on art-historical methodology and literary analysis, the article offers an overview of the many sorts of sources and materials that determine the perspectives we have of Daoism-related beliefs and concepts during the late Chosŏn. In contrast to earlier interpretations of Daoist practices as exclusively expressing a desire to retreat from public life, the materials discussed in this article advance a more subtle understanding of the pervasiveness of Daoism in late Chosŏn society, ranging from Daoist divination texts and rituals at religious shrines to the construction of artificial mountains for theater performances and the establishment of government office gardens that served as conduits for spiritual rejuvenation and display of cultural cachet.

**Keywords:** Daoist visual culture; Korean Daoism; Korean Buddhism; Chosŏn dynasty; Kitchen God; Sansin; Sinjung; Guan Yu; Guan Di; Kwanje; Jade Pivot Scripture; inner alchemy; islands of immortals; Mt. Penglai; Mt. Kŭmgang; Korean garden culture

## 1. Introduction

This study aims to bridge the divide between those who dismiss Daoism entirely in Chosŏn period Korea (1392–1910) and those who regard it as a third full-fledged religious tradition, alongside Confucianism and Buddhism. Writing a history of Korean Daoist visual and material culture is further complicated by the fact that there is as yet no comprehensive history of Daoism in Korea. Because this article covers a broad time period and a wide range of materials, I had to forego some of the finer-grained analysis that would be involved with studying a single site or a specific aspect of Chosŏn Daoism. Consequently, while this initial survey of material will present particular practices such as inner alchemy or Guandi 關帝 worship, future research will necessitate corrections and refinements of my account. But I hope that with this article the reader will become aware of the various sorts of sources and materials that determine the perspectives we have of Daoism-related beliefs and ideas in Chosŏn Korea.

This article illustrates the significance of Daoism in Chosŏn literati circles and Chosŏn society at large. Drawing from a wide range of sources, and not limiting itself to texts, this study argues for the need to recover Daoist concepts, visual traditions, and practices from the outdated interpretive habit of seeing them as politically motivated symbolical statements (i.e., ostentatiously withdrawing from a political world one disagrees with). Late Chosŏn Koreans worshipped Daoist icons, studied Daoist texts, and associated certain places, objects, and performances with otherworldly realms. Literate individuals associated with Daoist themes the idea that spiritual sensation could be attained by using appropriate material surroundings. Thus, in contrast to previous interpretations of Daoist practices as exclusively expressing a desire to retreat from public life, the materials discussed in this article offer an expanded understanding of the strongly spiritual as well as cultural and social aggrandizing aspects of Chosŏn Daoism.

But first, a word of caution on the term "Daoism." In a process comparable to the European fashioning of Buddhism as a philosophy rather than a religion, the Western imagination of Daoism has created two Daoisms, imposing an artificial separation between the "pure" (philosophy) and the "impure" (Daoist religious activities) (Robson 2015, p. 1474). To avoid such modern categorizations, I will use the term Daoism to refer to mainstream Sinitic religious practices as well as intellectual treatises. In contrast to the enormous Anglophone literature on Daoism in China, few scholars have examined Korean Daoism. Their contributions range from explorations of Daoist philosophical treatises and their impact on Chosŏn dynasty Neo-Confucian thinkers and Chinese medical knowledge to investigations into the political application of the Queen Mother of the West motif in late Chosŏn period royal court paintings.[1] This study contributes to the field by providing new insights into the functions of Daoism in the late Chosŏn, thereby adding to the budding field of Daoist visual and material culture in the broader, East Asian context.[2]

## 2. Daoist Gods and Ritual Texts in Late Chosŏn Korea

There is no recorded evidence that a nation-wide, systematized structure with Daoist temples and priests ever existed in Korea. The reason for this absence is that in Korea, Buddhist monasteries and shamanic shrines, as well as state shrines and private shrines, generally filled the niche for the worship of popular deities. Buddhist monks and shamans enshrined images representing gods that have since been classified as representations of Buddhist, Daoist, and ancient Indian beliefs. In order to gain a more nuanced understanding of Chosŏn Daoism, particularly when it comes to the relationship between Daoist, Buddhist and shamanic practices, I will begin with the analysis of various visual traditions related to the worship of Daoist icons, thereby complementing the scholarship on Korean Daoism which has so far largely relied on the analysis of literary texts.

Rare written material suggests that deified Chinese historical figures were worshipped at shamanic shrines in the mountains of Chosŏn Korea. A seventeenth century traveler's account includes an odd reference to Daoist/shamanic shrines at Mt. Kam'ak (Kamaksan 紺岳山) in P'aju 坡州, present-day Kyŏnggi Province 京畿道 (Kyŏnggido). While climbing the mountain, Hŏ Mok 許穆 (1595–1682) encountered a shrine dedicated to deified general Xue Rengui 薛仁貴 (6th–7th century), believed to be Mt. Kam'ak's mountain god (Sansin 山神). Hŏ Mok also saw a cave where a sculpture of Laozi was enshrined.[3] Such shrines were most likely managed by local shamans, Buddhist monks, and/or village elders rather than Daoist priests.

Mountain gods and other indigenous and foreign deities were also worshipped at Buddhist monasteries, the majority of which were likewise located in the mountains of the Korean peninsula. The main hall of a late Chosŏn Buddhist monastery typically had a painting of an assembly of guardian deities (*sinjung* 神衆) enshrined at the Western altar, which was dedicated to the lowest-ranking deities of the Buddhist pantheon.[4] The pantheon of guardian deities was initially based on the divine assembly of bodhisattvas, gods and supernatural beings that attended the Buddha's teaching of the *Flower Garland Sutra* (*Hwaŏm kyŏng* 華嚴經).[5] During the Three Kingdoms period (ca. 300–668 CE), Later Silla kingdom (668–935), and Koryŏ dynasty (918–1392), these protective deities appeared separately or in groups as sculpted images, and were represented in temple murals or on the exterior tiles of granite stone pagodas. Their main function was to protect the ruling house and country from evil influences and foreign invasions. The mode of depiction and the roles of these deities shifted during the early Chosŏn in response to changing ritual protocols which included the Water-Land-Assembly (*Suryukchae* 水陸齋). Protective deities began to be shown as a large group on separately crafted banner paintings, with an emphasis placed on their roles as protecting individual devotees and temples from calamities and disasters, as well as exorcising evil spirits (Kim 1997a, pp. 211–16). These large paintings featured the originally Vedic deity Skanda (Wit'aech'ŏn 韋馱天), who was considered protector of the *dharma*, destroyer of all evil, and leader of ghosts, as well as esoteric deities like Ucchuṣma (Yejŏkkŭmgang 穢迹金剛), indigenous Sinitic deities like the Kitchen God

(Chowang 竈王, C. Zaowang), and diverse local Korean deities including the Mountain God.[6]

A mid-nineteenth century *sinjung* painting enshrined in the Vairocana Hall (Taejŏkkwangjŏn 大寂光殿) of Haein Monastery 海印寺 is a representative example for an expanded pantheon with 124 guardian deities (Figure 1) (Kim 1997a, p. 222; 1997b, p. 233; 239 ff). The painting was created by a group of itinerant painter-monks led by Tŏgun 德芸 who was active primarily in the Yŏngnam 嶺南 region where Haein Monastery is located. The composition is split into three sections, while each corner of the composition is guarded by one of the Four Heavenly Kings (Sach'ŏnwang 四天王). Nine figures with malachite green nimbi are depicted in larger scale than the various *devas*, *nagās*, eight kinds of beings (*ch'ŏllyong p'albujung* 天龍八部衆), and celestial musicians that surround them. The upper section is centered on the Vedic gods Śakro devānām indraḥ (Chesŏk 帝釋, abbreviated Sanskrit term: Indra) on the left and Brahmā (Pŏmch'ŏn 梵天) on the right, flanked by the solar and lunar deities ( Ilch'ŏn 日天; Wŏlch'ŏn 月天). The Dragon King (Yongwang 龍王), standing next to the lunar deity, can be identified by his white hair, spikey coral-shaped eyebrows and spikey beard. The central section is framed by four haloed bodhisattvas, who accompany the three-eyed and eight-armed Maheśvara (Taejajaech'ŏn 大自在天) on the right, and three-headed Ucchuṣma on the left. The painter-monks highlighted Ucchuṣma with dramatically upturned eyes, fire-spitting lips, and grasping the moon and a trident in two of his hands. Eight Vajra Warriors (Kŭmgangyŏksa 金剛力士) wield their swords between these two deities, while ten Wise Kings (Myŏngwang 明王) appear below them, carrying jade tablets in their hands. The lower section depicts a swarm of fierce-looking eight kinds of beings (i.e., *devas*, *nāgas*, *yakṣas*, *gandharvas*, *asuras*, *garuḍa*, *kiṃnara*, and *mahoraga*) clad in armor, led by Skanda who brandishes a spear and wears a winged helmet embellished with bird feathers. Skanda is flanked by the white-bearded Mountain God who is clutching a fan, the Kitchen God in his official robe and crown, and Guan Yu, who is identifiable by his dark-red visage, three-pronged dark beard, golden crown, and long blue dragon sword.[7] These *sinjung* were not only summoned during nearly every ritual in the temple's main hall, but they were also called upon during special events such as the *Suryukchae* to deliver creatures of water and land, and New Year celebrations to safeguard Buddhist monasteries from disasters in the upcoming year.[8]

*Sinjung* paintings such as the above-discussed *124 Guardian Deities* provide important evidence for the wide range of guardian deities in the late Chosŏn Buddhist context. If one pursues a more traditional binary framework to the study of Buddhist deities and "foreign" deities such as Vedic deities, Buddhist esoteric deities, or Daoist deities, one might contrast the differences in their meaning and function depending on the religious context, for example the invocation of Ucchuṣma in Buddhist protection rituals versus Daoist exorcistic rituals; or the apotropaic function of the Kitchen God depicted in a *sinjung* painting versus the Daoist interpretation of the Kitchen God as a messenger who reported good and evil deeds of family members to the Jade Emperor. Another strategy would be to recognize the interconnectivity between pantheons of many belief systems, such as the Vedic tradition as well as Daoism, Buddhism, Neo-Confucianism, and local religious practices, and to acknowledge the flexibility of these deities' identities. Thus, *sinjung* paintings wonderfully illustrate the convergence of multiple cultural and religious elements in the Chosŏn religious landscape.[9]

Worship of Guandi 關帝 (Kwanje; also Kwanwang 關王; Muanwang 武安王), the apotheosized version of the third-century Chinese general Guan Yu 關羽 (160–220 CE), is another splendid example of a network of Chosŏn religious beliefs in both the official and private contexts. Despite the fact that Luo Guanzhong's 羅貫中 (1320–1400 CE) early Ming novel *Romance of the Three Kingdoms* (C. *Sanguo yanyi* 三國演義, also called *Sanguozhi yanyi* 三國志演義) introduced Guan Yu to early Chosŏn literate circles, Ming generals are usually credited for bringing this deity to Korea during the Imjin War (1592–98 CE, also referred to as Hideyoshi invasions). Royal records deliberately imply that, although initially opposed to the worship of a martial deity, Neo-Confucian authorities eventually justified

Kwanje worship by arguing for the need to commemorate heroes of the (Chinese) Three Kingdoms period.[10]

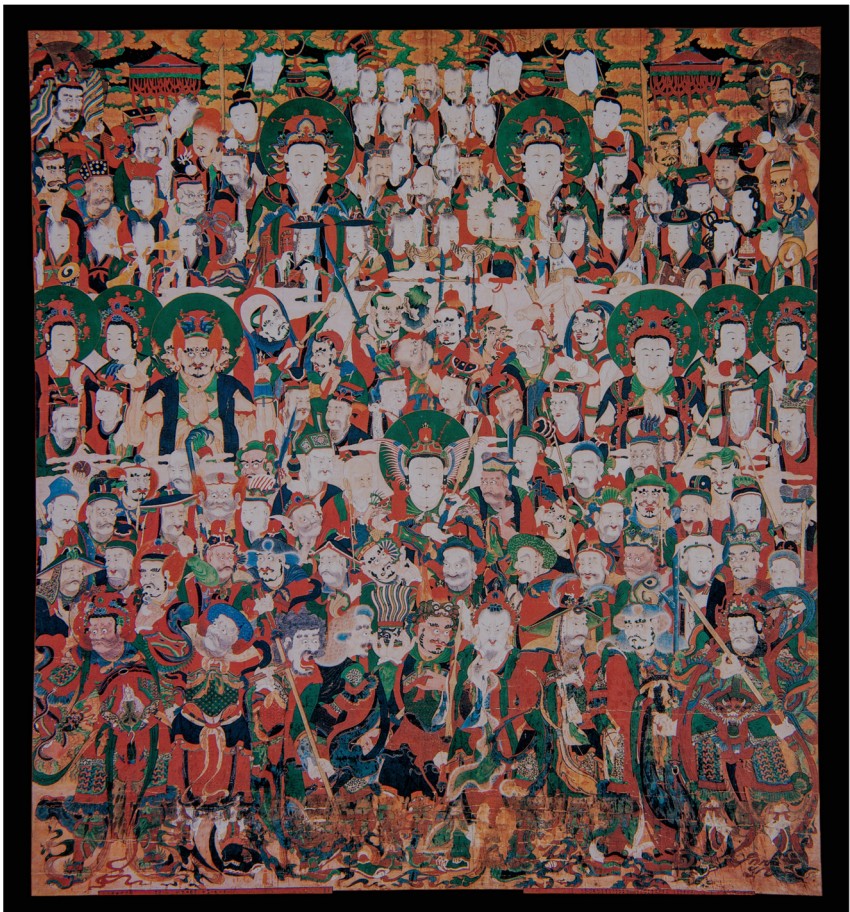

**Figure 1.** Tŏgun et al. *124 Guardian Deities*, 1862, Chosŏn dynasty (1392–1910). Colors on silk, 348 × 315.5 cm. Haein Monastery, Kyŏngsang Province. Reproduced with permission from the Research Institute of Sungbo Cultural Heritage, Seoul.

Beginning with King Sŏnjo's 宣祖 (1552–1608, r. 1567–1608) reign, Kwanje was venerated at state-sponsored shrines including the Southern Shrine (Nammyo 南廟) and Eastern Shrine (Tongmyo 東廟) in Seoul. These shrines were accessible to the public, since visiting the shrine of the paragon of filial piety and loyalty was intended to raise the morale among the populace. However, shrine visitors not only worshipped Kwanje but also took a piece of his beard, presumably for talismanic purposes, with the effect that the sculpture's long beard had become suspiciously short when King Sukchong visited one of the shrines in 1691—the King promptly ordered the restoration of Kwanje statues at the Southern and Eastern Shrines.[11] The Kwanje sculpture enshrined in the main hall of the Eastern Shrine today is most likely one of the restored images (Figure 2; cf. National Museum of Korea 2013, pp. 238–39). It depicts the warrior god in Ming-style armor and with a moustache and long beard. According to the construction record it was originally created by a group of Ming Chinese and Chosŏn Korean sculptors in 1601. The creation of this 250 cm (ca. 8 feet) tall sculpture required several attempts and raw material in the amount of 2.4 tons of copper, which were provided to the sculptors with support of the Ming and Chosŏn government (Chang 2013, p. 100 ff.). The brief episode discussed above suggests that the Kwanje cult had become established in both royal ritual and popular religious context by the seventeenth century at the latest, likely facilitated by earlier popular versions of this cult and folktales rather than the novel *Romance of the Three Kingdoms* (cf. Walraven 2015, p. 217; Ter Haar 2017, p. 16).

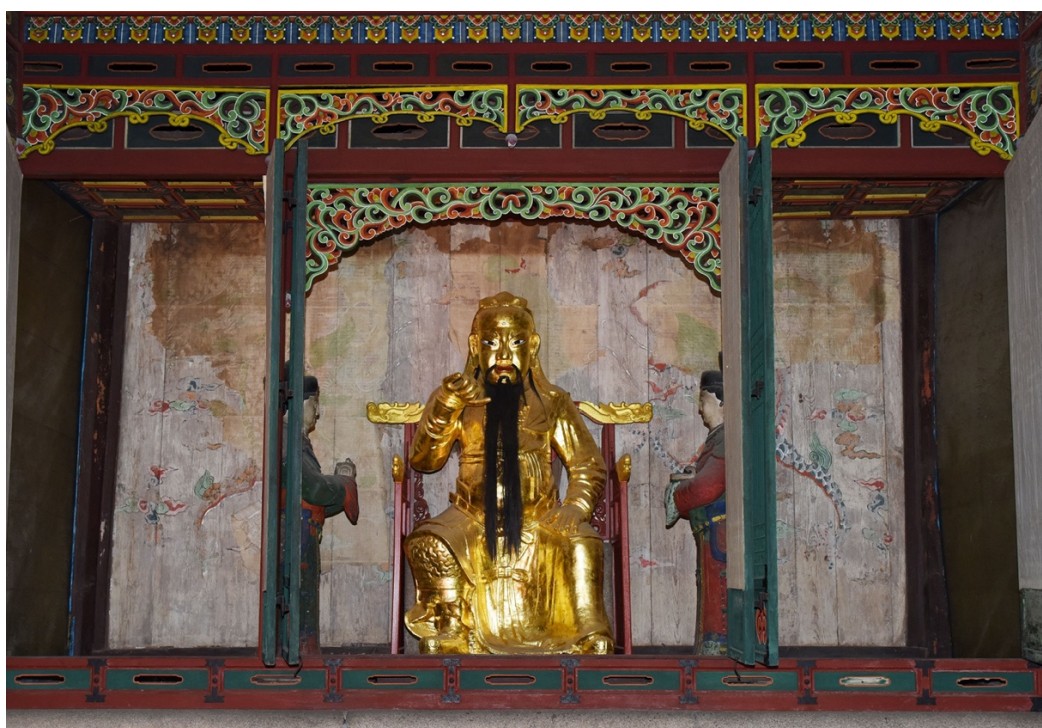

**Figure 2.** *Kwanje (C. Guandi) with two female attendants*, 1601. Main icon: gilt bronze, H 250 cm. Attendant figures: mineral colors on clay, H 187 cm. Main Hall of Eastern Shrine, Seoul. Image licensed under Creative Commons, reproduced with permission from the Korean Cultural Heritage Administration, Taejŏn.

Since Kwanje was a deity concerned with the defense of legitimate rulers and loyal subjects, his cult was politically beneficial for resolving issues of royal legitimacy and taming the ferocity of court factionalism from the 1680s onward, i.e., the reigns of kings Sukchong 肅宗 (1661–1720, r. 1674–1720), Yŏngjo 英祖 (1694–1776, r. 1724–1776), and Chŏngjo 正祖 (1752–1800, r. 1776–1800).[12] Through the worship of Kwanje, the Chosŏn kings were able to promote the ideal of his loyalty to the throne and martial valor among their subjects. What complicates the worship of Kwanje in Korea is the fact that he was depicted as a Chosŏn monarch from King Yŏngjo's reign onward, since Yŏngjo linked the loyalty of Guan Yu to Chosŏn's gratitude to the Ming for entering the peninsula during the Imjin War. Accordingly, Kwanje had evolved into a protector deity of the Chosŏn royal house by the early eighteenth century at the latest. The Kwanje statue at the state shrines came to be covered by a piece of real clothing, probably a king's daily robe known as the dragon robe (*kollyongp'o* 袞龍袍), to emphasize Kwanje's role as the Chosŏn state's tutelary guardian. Further evidence other than royally robed sculptures would be needed to substantiate this claim, but it almost seems as if late Chosŏn monarchs desired to be portrayed as a personification of Kwanje. King Chŏngjo furthered the promotion of the Kwanje cult, by having him portrayed as the embodiment of loyalty and martial spirit as well as the tutelary deity of Chosŏn in the lyrics of newly composed ritual music for the shrines (Lee 2020, p. 206).

The late eighteenth and nineteenth century iconography of Kwanje dictated a figure seated in frontal pose, with a fierce-looking face with a dark red complexion, a three-pronged beard, and two jade belts. The Northern Shrine's Kwanje sculpture, which had been commissioned by King Kojong 高宗 (1852–1919, r. 1907–1910) and Queen Min 閔 (1851–1895) in 1883, exemplifies this trend (Figure 3). It depicts a fierce looking Kwanje dressed in a royal dragon robe, seated in front of a symbolic landscape screen representing the universe. In the royal court context, the figure was further depicted with a winged coronet called *iksŏngwan* 翼善冠 and a red dragon robe, which sets the Korean iconography apart from late Chinese visual conventions that portrayed Guandi as a mighty warrior in

a green robe over armor with a dark hood (Figures 4 and 5). (Lee 2020, p. 204; Chang 2008, pp. 98–99).

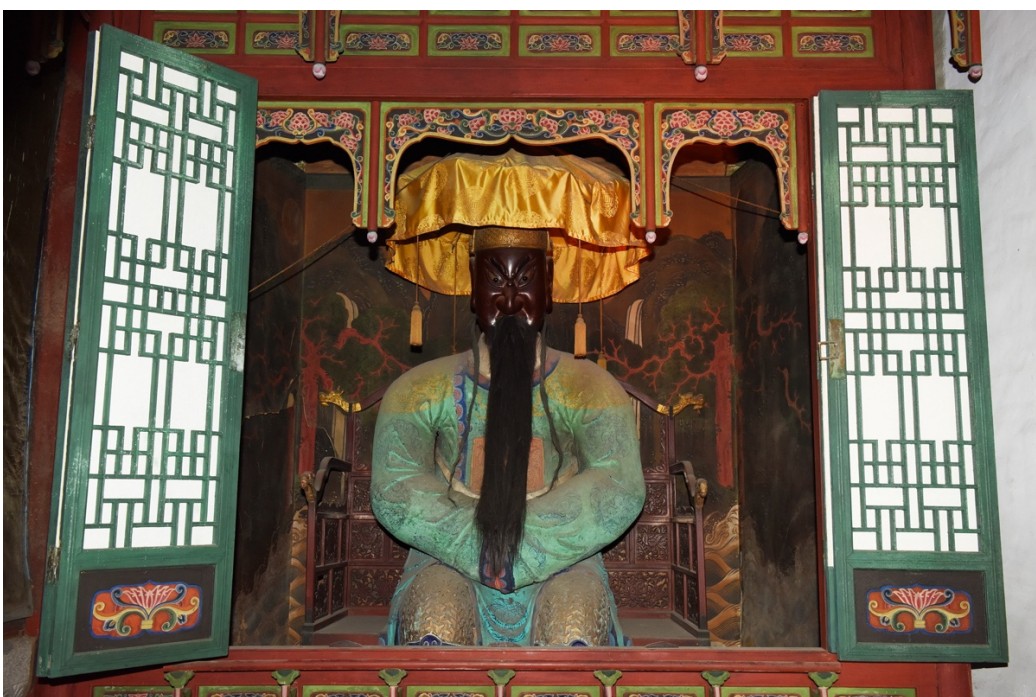

**Figure 3.** *Kwanje in dragon robe*, formerly main icon of the Northern Shrine, 1883. Red lacquer on wood (head) and molded clay (body). H 216 cm. Main Hall of Eastern Shrine, Seoul. Image licensed under Creative Commons, reproduced with permission from the Korean Cultural Heritage Administration, Taejŏn.

By the early twentieth century at the latest, Kwanje had developed into a major popular deity believed to cure sicknesses and bestow wealth. To combat "illicit cults" in Seoul in 1904, the police seized 3000 images of Kwanje from private residences. In addition to the official temples in Seoul, a Japanese researcher recorded 107 private shrines for Kwanje. Merchants in Seoul were among his most fervent worshippers and built small shrines for him near their markets. Guan Yu and other heroes from the *Romance of the Three Kingdoms* were also included in the pantheon of shamans (Walraven 2015, p. 217). Modern shamanic icons modeled after the traditional shamanic iconography of Kwanje accentuate his virility by showing him with a long, thick beard, a martial weapon, and headgear, but do not display any royal regalia (Figure 6). Kwanje's flexible iconography is a perfect illustration of the various ways in which the visual lexicon of imported Daoist deities was re-calibrated in the Korean religious and political realm.

But the influence of Sinitic Daoist traditions extended beyond visual conventions and the worship of icons. In the nineteenth century, Eastern Doctrine (Tonghak 東學, today known as the Teaching of the Heavenly Way, or Ch'ŏndoism, Ch'ŏngdogyo 天道敎) and Teaching of Chŭngsan (Chŭngsan'gyo 甑山敎; today known as Way of Chŭngsan, Chŭngsando 甑山道) were two new religions that blended Daoist concepts into their own teachings (Van Lieu 2019, p. 91ff; Jung 2000, pp. 801–2, 816; Kwon 2009). Furthermore, by publishing scriptures received via spirit-writing (*nansŏ* 鸞書), religious groups like the Formless Altar (Musangdan 無相壇) promulgated the teachings of the Three Sages, i.e., Guandi, Imperial Sovereign Wenchang (Wenchang dijun 文昌帝君), and Imperial Sovereign Fuyou (Fuyou dijun 孚佑帝君).[13]

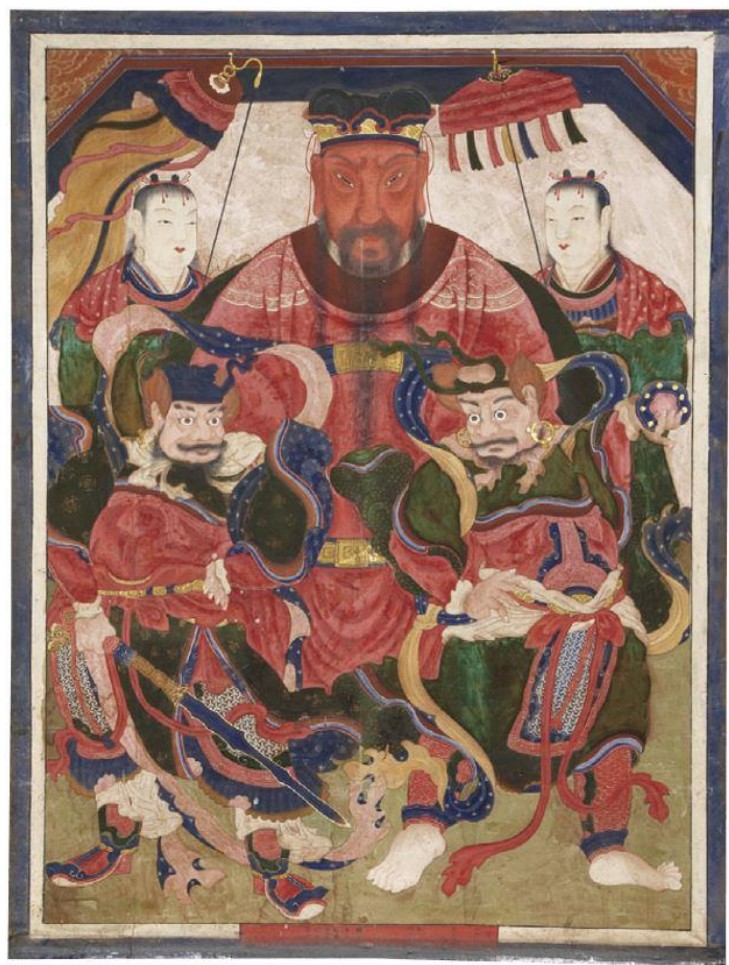

**Figure 4.** Taehŏ Chehun 大虛體勳 and Hakhŏ 鶴虛. *Kwanje/Temple God* (Toryangsin 道場神), 1885. Chosŏn dynasty (1392–1910). Mounted painting, colors on silk. 150.5 cm × 112.4 cm. Ultimate Bliss Hall, Hŭngch'ŏn Monastery, Seoul. Photograph by Chang Chi-hŭi, 2019. Image licensed under Creative Commons, reproduced with permission from the Korean Cultural Heritage Administration, Taejŏn and the Buddhist Research Institute of Cultural Heritage, Seoul.

The use of Daoist texts for ritual practices is another factor that provides a richer and more nuanced understanding of late Chosŏn Daoism. The *Yushu baojing* 玉樞寶經 (Precious Scripture of the Jade Pivot, K. *Okchu pogyŏng*), also known by its full name, *Jiutian yingyuan leisheng puhua tianzun yushu baojing* 九天應元雷聲普化天尊玉樞寶經 (Precious Scripture of the Jade Pivot, Spoken by the Celestial Worthy of Universal Transformation of the Sound of the Thunder of Responding Origin in the Nine Heavens), is a beautifully illustrated text that exemplifies these practices. This text, which in Imperial China served as the foundation for the Thunder Rites of the Shenxiao 神霄 tradition, was attributed to the Supreme God of Thunder (Puhua tianzun 普化天尊), the Daoist form of the Bodhisattva Samantabhadra (K. Pohyŏn posal 普賢菩薩).[14]

Three annotated Chosŏn editions have so far been found in South Korean and Japanese library, museum, and temple collections. The Mt. Mudŭng Ansim Monastery (Mudŭngsan Ansimsa 無等山 安心寺) edition, printed in 1570, is the oldest; followed by the Mt. Myohyang Pohyŏn Monastery (Myohyangsan Pohyŏnsa 妙香山 普賢寺) version edition produced in 1733, and the Mt. Kyeryong (Kyeryongsan 鷄龍山) edition, printed in 1888.[15] With the exception of the absence of a frontispiece in the Korean versions, the Chosŏn editions have a comparable pictorial arrangement that appears to reflect earlier Yuan and Ming editions like the copy in the British Library's collection (acc. no. 15103.aa.2) (Little and Eichman 2000, pp. 237–39; Wan 2010). A total of 33 illustrations depict a Daoist mas-

ter's visualization journey to heaven, followed by eight pages depicting talismanic seals that might have been written or used in the visualization. The postscripts list the names of several dozen donors (mostly laypeople), fundraisers, carvers, as well as the print's production date and location.

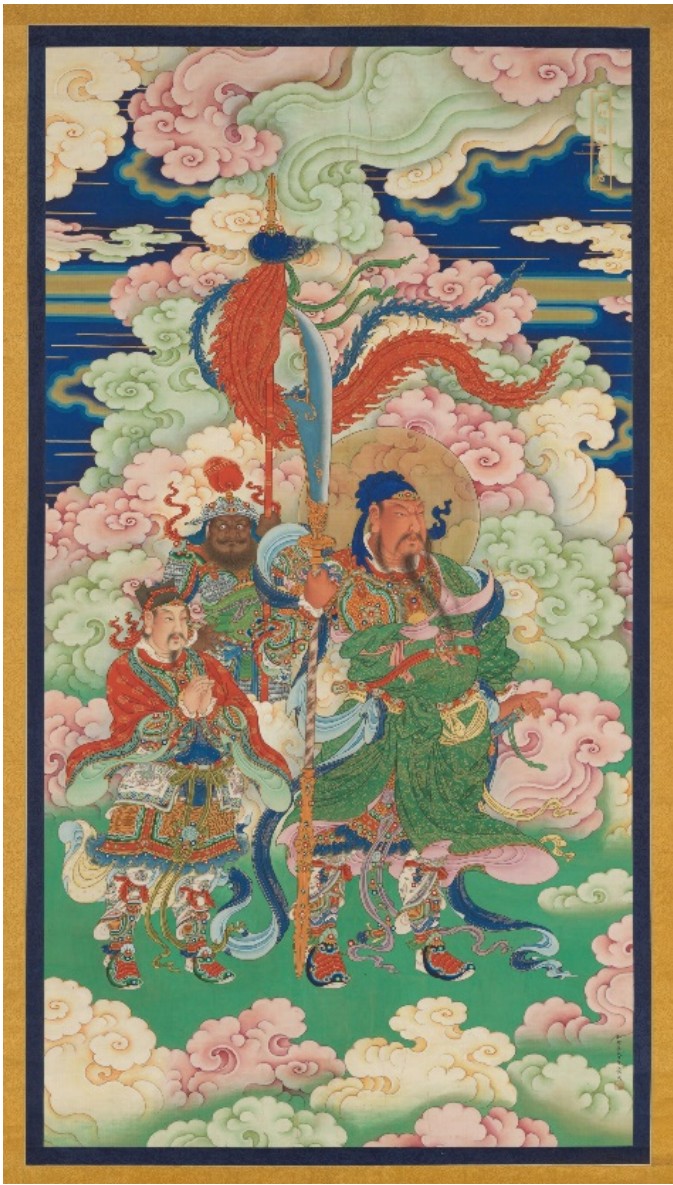

**Figure 5.** Unidentified artist. *Guandi*, ca. 1700. Qing dynasty (1644–1911). Hanging scroll (from a set of images for *shuilu* 水陸 rituals); ink, color, and gold on silk. 173 cm × 92.6 cm. Metropolitan Museum of Art (2001.442), purchased with the B. Y. Lam Fund and Friends of Asian Art Gifts, in honor of Douglas Dillon, 2001. Image licensed under Creative Commons, reproduced with permission from the Metropolitan Museum of Art, New York.

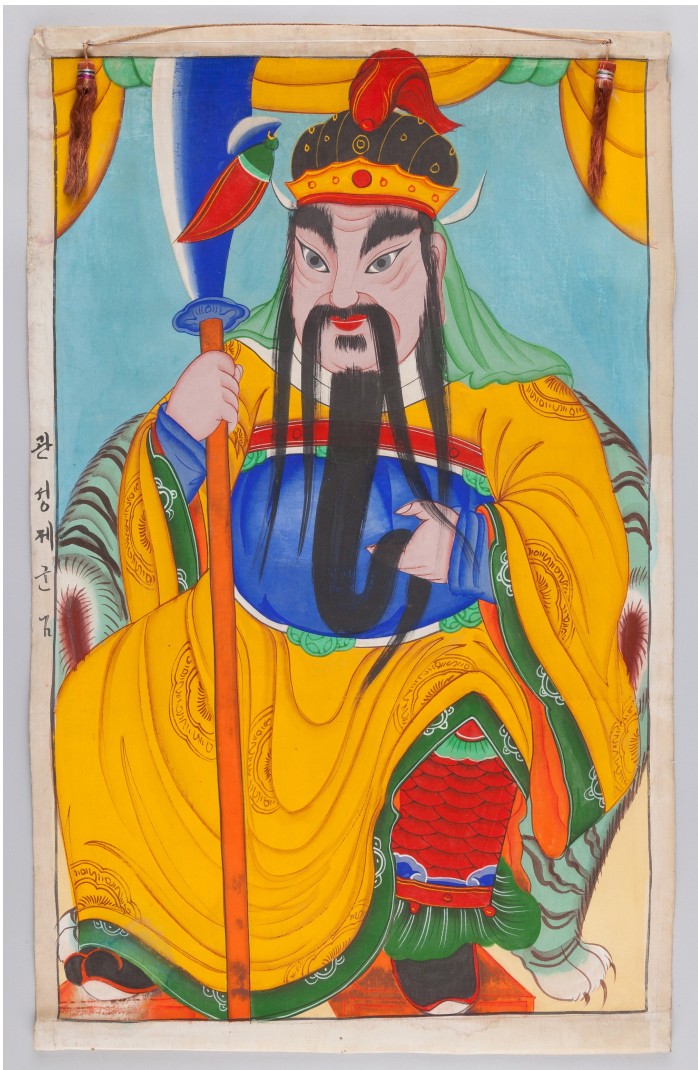

**Figure 6.** Hwang Ch'un-sŏng. *Emperor Guan* (Kwansŏng chegundo 關聖帝君圖). Post-1945, S. Korea. Colors on silk. 83.5 cm × 53 cm. National Folk Museum of Korea, Seoul (minsok 072639). Image licensed under Creative Commons, reproduced with permission from the National Folk Museum of Korea, Seoul.

Like the other surviving versions, the final illustration of the 1570 Ansim Monastery version (Figure 7), depicts the most powerful visualization of the Supreme God of Thunder's martial form, riding a *qilin* galloping through clouds and seas. A Daoist high god identifiable as the Highest Prince of Jade Purity (C. Wushang yuqing wang 無上玉清王) is depicted in the upper right corner of the pictorial composition, sending a beam of energy to the Supreme God of Thunder from his raised hand. Numerous thunder gods and officials are joining the scene in the upper left. Four figures in charge of punishing the wrongdoers are depicted in the lower left corner, while a figure carrying the records of good and bad actions is shown below the Supreme God of Thunder, directing his way. Given how well this image was rendered in great detail, in comparison to the preceding illustrations, it must have held exceptional significance as an effective visual and ritual instrument for accessing the Supreme God of Thunder's power. When creating talismans that were believed to cure ailments and keep people safe from harm, Buddhist monks, blind diviners, and shamans probably visualized the Supreme God of Thunder while chanting this text. Laypeople supposedly also chanted this scripture for a variety of reasons, including the treatment of illnesses.[16]

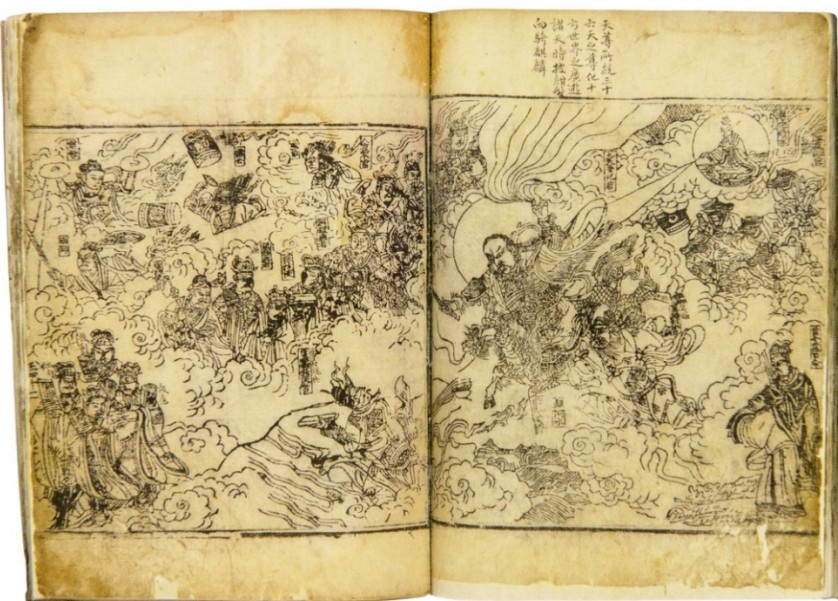

**Figure 7.** Illustration from the Ansim Monastery print of the annotated *Yushu baojing*, 1570, Chosŏn dynasty. Booklet, illustrated woodblock print. 33.8 cm × 26.1 cm. Collection of Woodblock Print Museum, Wŏnju, Kangwŏn Province, South Korea. Reproduced with permission from Woodblock Print Museum.

There is a large corpus of hand-written and printed texts on inner alchemy (*naedan* 內丹) that have yet to be properly investigated by contemporary scholars.[17] In his *Commentary to the Cantong qi* (*Ch'amdonggye chuhae* 參同契 註解), Kwŏn Kŭk-chung 權克中 (1585–1659), established a systematic inner alchemy philosophy that encompasses a complete ontology, theory of human nature, system of alchemical practice, and doctrine of immortality. Sin Ton-bok's 辛敦復 (1692–1779) *Mirror of Instructions and Reflection on Daoism* (*Toga chikchi tokcho kyŏng* 道家直指獨照經) is an excellent example for late Chosŏn literati's engagement with Daoist doctrine. It includes a collection of the principles of moral enlightenment required before practicing inner alchemy, and main themes of spiritual and physical discipline, from practical plans for physical training to self-cultivation through taking herbs and regulating the interior fire. Kang Hŏn-gyu 姜獻奎 (1797–1860) authored a book titled *Interpretation of the Cantong qi* (*Chuyŏk ch'amdonggye yŏnsŏl* 周易 參同契演說, abbreviated *Interpretation*), which was first published in 1857.[18] The title is deceptive because it is not a commentary on the *Cantong qi*, but rather a compilation of significant Daoist writings that systematically consolidates selections from Korean and Chinese texts dealing with inner alchemy, cultivation, breathing exercises, gymnastics, and morality (Figure 8) (Jung 2000, pp. 805–9).

In his personal writings, Kang describes a spiritual experience that suggests his physical engagement with Daoist practices prior to the publication of *Interpretation*. When visiting the Tower of Divine Transcendent Beings (Sinsŏndae 神仙樓) at Changan Monastery 長安寺 (Changan Monastery) in Mt. Kŭmgang, a place believed to be the playground of transcendents, Kang's exposure to the marvelous scenery of mountain peaks unfolding in front of his eyes triggered the feeling of being light as a feather ascending to heaven, i.e., becoming a transcendent being.[19] Such writings and recorded experiences constitute important evidence for late Chosŏn scholarly interest and engagement with Daoist material. The Chosŏn state obviously tolerated such elite engagement with Daoism as long as Neo-Confucian orthodoxy was maintained in the public realm.

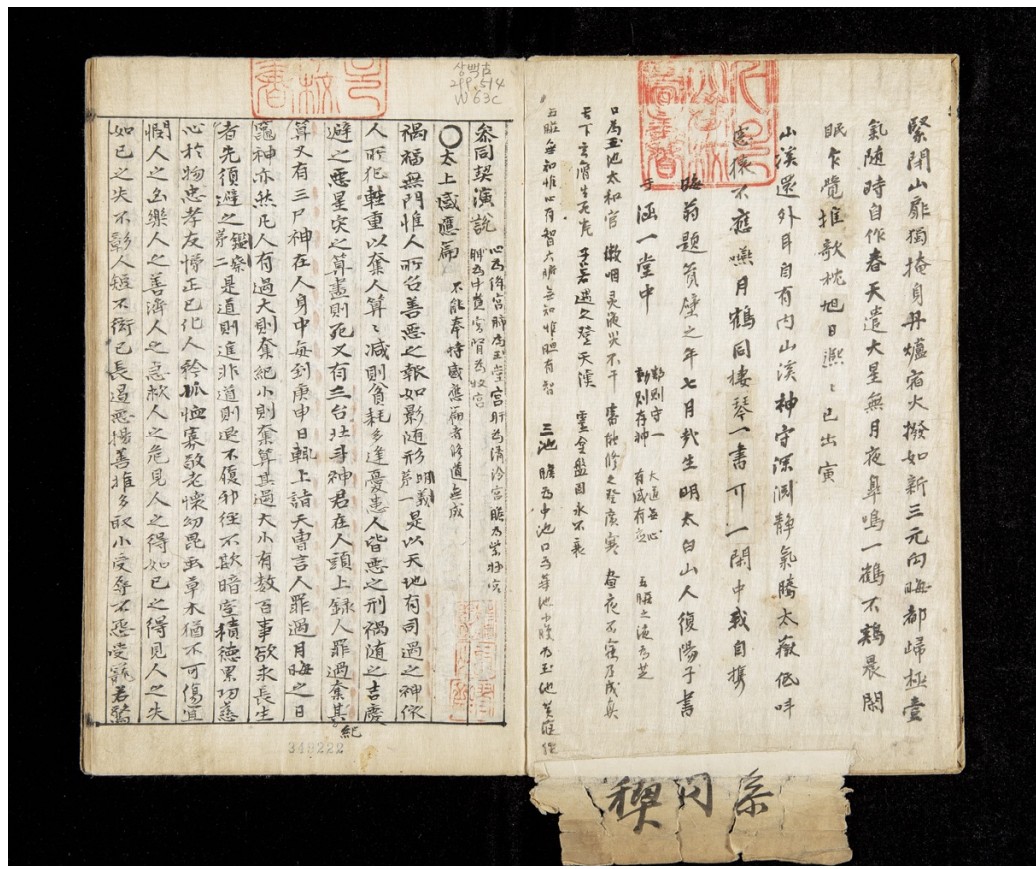

**Figure 8.** Kang Hŏn-gyu, *Interpretation of the Cantong qi*, 1857, Chosŏn dynasty (1392–1910). Ink on paper. Kyujanggak Institute of Korean Studies/Seoul National University Central Library ( 想白古 299.514-W63c). Reproduced with permission from Kyujanggak Institute of Korean Studies/Seoul National University Central Library.

## 3. Representations of Otherworldly Mountains in Late Chosŏn Culture

An integral part of visual, material, and performative culture, representations of otherworldly mountains vividly demonstrate that the concept of temporarily withdrawing to an idealistic place was a common phenomenon in late Chosŏn culture and was not just used as an expression of political aversion. A water dropper on one's desk or a theater stage; actual landscape infused with Daoist mythology, or a garden inspired by the three islands of immortality, these are just a few examples of otherworldly things and places that late Chosŏn Koreans encountered or even created.

One kind of tangible object used to portray an otherworldly land was a water dropper shaped like a universal mountain *(paksan* 博山). Such water droppers probably allowed late Chosŏn cultured individuals to experience a brief spiritual escape while engaging in literary pursuits. The National Museum of Korea has an exceptional example of such a water dropper in their collection (Figure 9). Dated to the nineteenth century, it was made from high-quality porcelain, shaped like a steep cone, with cobalt blue and copper red painting in underglaze accentuating the tips of the peaks. According to the current scholarship, this type of water dropper is believed to represent Kŭmgangsan. However, due to its formal features it seems more plausible to me that it is a small representation of an unspecified otherworldly mountain with transcendent beings standing between and on top of its pointy, triangular peaks, similar in shape to the ancient Chinese form known as *boshanlu* 博山爐 (universal mountain incense burner), and reminiscent of Korean depictions of mythical landscapes such as An Kyŏn's 安堅 (act. 15th century) *Painting of a Dream Journey to the Peach Blossom Land* (*Mongyudowŏndo* 夢遊桃源圖) (cf. Lee 2018, p. 130).

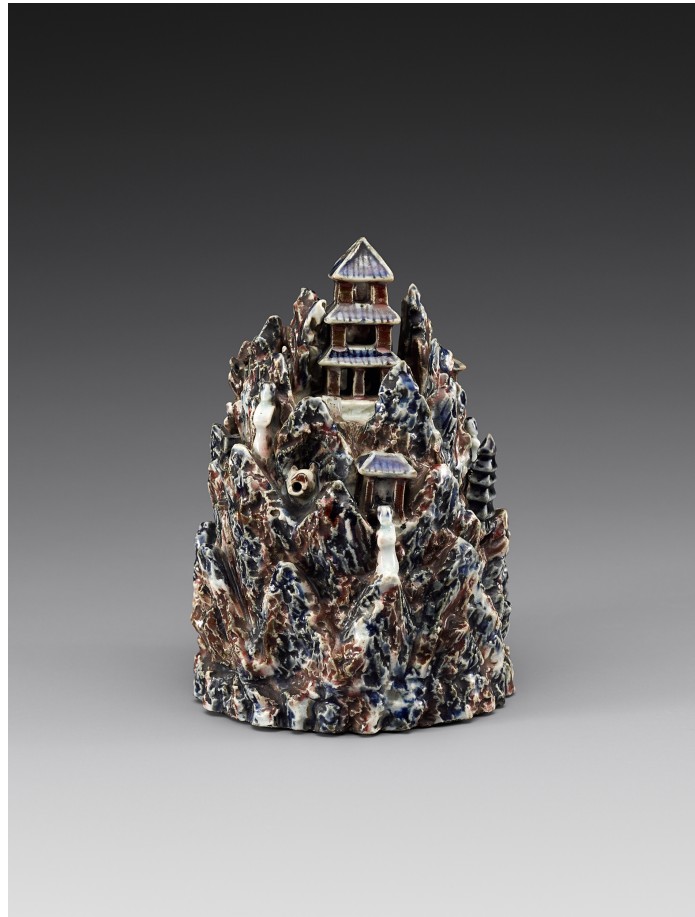

**Figure 9.** *Paksan*-shaped Water Dropper, 19th century, Chosŏn dynasty (1392–1910). Porcelain with cobalt-blue and copper-red painting in underglaze. H 19.0 cm. National Museum of Korea (pon 8143). Photograph licensed under Creative Commons, reproduced with permission from National Museum of Korea, Seoul.

Daoist concepts also had vernacular amplifications, such as puppet theater performances like *Mt. Kŭmgang Performance* (*Kŭmgangsan nori*), *Divine Transcendents' Performance* (*Sinsŏn nori*), and *Eight Female Celestials' Performance* (*P'alsŏnnyŏ nori*), which were performed with a *yesandae* 曳山臺 (also known as *sandae* 山臺).[20] A *yesandae* was a vertical, mountain-shaped stage mounted on a cart. It measured several dozen feet and was made from bamboo poles, cotton fabric, and clay. The above-mentioned plays belonged to the repertoire of *sandae togam* 山臺都監, a government-supported troupe particularly known for their *sandae* performances. They organized and executed grand-scale performances at the royal court and the streets of Seoul at the end of each year. The royal court also mobilized *sandae* performers for the entertainment of high-ranking foreign envoys, but since stage preparation for *sandae togam* required several dozen to several hundred workers and a considerable amount of material, royal support dwindled in the late eighteenth century.[21] In late nineteenth-century Seoul, *sandae togam* only performed on rare special occasions, for example, when the king visited the reconstruction site of Kyŏngbok Palace (Kyŏngbokkung 景福宮) in 1865. On the day of the occasion, the anonymous author of the *Special Record about Odd Amusements* (*Kiwan pyŏllok* 奇玩別錄) witnessed a *sandae* street performance in front of the royal palace gate Kwanghwamun 光化門 where *sandae togam* had traditionally performed for the royal court and its visitors since the early Chosŏn (Sa 1998, pp. 354–57; 2002, pp. 348–49, 411).

In one of his paintings from the album *Illustrations of an Imperial Commissioner* (Fengshitu 奉使圖), Akedun 阿克敦 (1685–1756), a Qing (1636–1911) Chinese envoy who visited

Korea in 1725, depicts a *sandae* stage (Figure 10) (Chŏng 2005, p. 218). Based on sketches he commissioned from a Korean artist, Akedun's detailed depiction of the scene captures the moment when two men (possibly members of the *sandae togam* troupe) wheeled a several-dozen-feet tall, bizarrely shaped artificial rock mountain to the outdoor stage. Niches inside the rock feature painted clay figurines of a female wearing a fancy, bright red skirt, as well as a monkey and an angler. Green-leaved twigs and branches adorn the rock, evoking the natural characteristics of mountain scenery that suggests an imaginary otherworldly land. While the *Special Record about Odd Amusements* refers to *sandae* as a representation of Mt. Kŭmgang that was used as a stage to perform the tale of Kim Man-jung's 金萬重 (1637–1692) *Cloud Dream of the Nine* (*Kuunmong* 九雲夢), Akedun's album leaf demonstrates the use of *sandae* as a backdrop which enhanced the visual impact of various performances that took place in front of it, including mask dance, plate spinning, and tightrope walking. Framed by small groups of onlookers of varying age, gender and social status, the scene contributes to our understanding of the extent to which Daoist-(and Buddhist-)infused performance arts attracted a wide range of people in the late Chosŏn (cf. Sa 2002, pp. 358–61, 374).

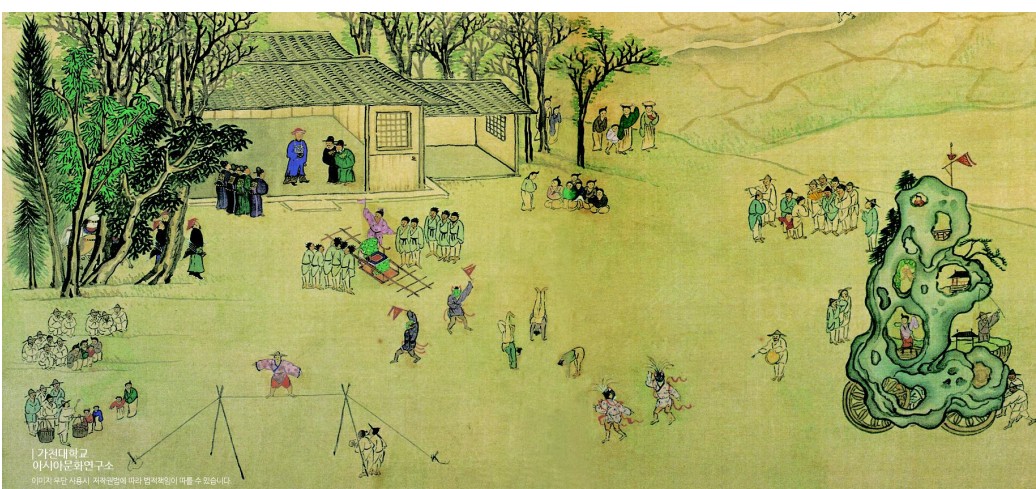

**Figure 10.** Akedun (1685–1756), Leaf no. 7 from the album *Illustrations of an Imperial Commissioner*, 1725, Qing dynasty. Ink and colors on silk, 40 cm × 51 cm. National Library of China, Beijing. Reproduced with permission from Research Institute of Asian Cultures, Gachon University, Sŏngnam.

## 4. Conceptualizing Korean Mountains as Otherworldly Lands

Due to insufficient source material we know very little about popular references to transcendents and Daoist mythology other than the theater stage discussed above. But Daoist references abound in Chosŏn literary writings, which serve as a vital resource for today's Korean studies researchers. Late Chosŏn cultured individuals designated worthy and recognizable places to position themselves on the grid of Sinitic literati culture, in which naming and validating place names was an essential part of travel behavior.[22] During their journeys, late Chosŏn travelers noted natural formations such as rocks or boulders, as well as human-made bridges and pavilions. When Yi Myŏng-jun 李命俊 (1572–1630) traveled to Mt. Kŭmgang in 1628, he named a large rock the Boulder of Five Transcendent Beings (Osŏn'am 五仙巖) and had its name carved into it to ensure that future visitors would recognize and use the name he had given to the boulder. Kim Su-jŭng 金壽增 (1624–1701) followed a similar strategy when he traveled to Mt. Kŭmgang in 1680, and had the name Bridge Where One Asks About Transcendent Beings (Munsŏn'gyo 問仙橋) carved into a boulder adjacent to a freshly restored bridge at Changan Monastery. What is most intriguing about Yi's and Kim's naming practices is that the concept of immortal lands was central to the meaning of their place names and to their vision (or hope?) of transformation into a transcendent being.

Chosŏn literati frequently allude to otherworldly lands and the concrete notion of Daoist self-cultivation in their travel accounts. Sŏng Che-wŏn 成悌元 (1506–1559) recounts the spiritual experience of becoming an immortal in his journal Record of Traveling to Mt. Kŭmgang (*Yu Kŭmgangnok* 遊金剛錄):

> "Litting the lamp and lying down by myself, I pondered about the trip. My mind was clear, as if I had ascended to the purple [emperor's] kingdom. I had shed the bark and transformed into a transcendent being."[23]

The orthodox Confucian scholar Yi Ch'ŏn-sang 李天相 (1637–1708) had a similar experience during his trip to Mt. Kŭmgang in 1672. Yi was a local aristocrat from the Yŏngnam area. He never took the government exams but wrote several books on Neo-Confucian thought, such as the *Neo-Confucianism Guidebook* (*Sŏngni chinam* 性理指南). While enjoying the grand view from Heavenly Escape Terrace (Ch'ŏnildae 天逸臺) at P'yohun Monastery (P'yohunsa 表訓寺), he accounts the following:

> "Suddenly my body was riding a crane and I was without a single notion of dirt, as if I went up to heaven and acquired the ability to ride the wind, as if I did not want to return to my former appearance."[24]

While Sŏng Che-wŏn and Yi Ch'ŏn-sang utilize Daoist nomenclature to describe a personal spiritual experience they had while climbing Mt. Kŭmgang, Yi I 李珥 (1536–1584) equates the mountain with the land of immortals in his Record of P'ungak (*P'ungaknok* 楓嶽錄):

> "Men with wings live at this mountain
>
> Riding the wind and flying in the air
>
> Living for one thousand years by eating pine needle oil
>
> Attaining longevity by casting off the skin [of the mundane world]."[25]

Since the early days of the Chosŏn, cultured individuals not only recorded their spiritual experiences at the mountain but also referred to Mt. Kŭmgang as the imaginary Mt. Penglai (C. Penglaishan 蓬萊山), interpreting the desolate area of Kwandong 關東 in Kangwŏn Province (Kangwŏndo 江原道), where the mountain is located, as an otherworldly place. I argue that in doing so, the literati transformed a local mountain into an ideal place for spiritual purification. As seen by the records of Yi Ch'ŏn-sang and Yi I, scholars strictly adhering to Neo-Confucian orthodoxy were thereby able to legitimize journeys to Mt. Kŭmgang which had historically strong Buddhist connotations (Stiller 2021, pp. VII–X).

So far, above-discussed poems have been taken as evidence of mid-Chosŏn literati's yearning to withdraw from society. Indeed, the mid-Chosŏn was a time marked by literati purges, the emergence of factionalism at the royal court, the devastating invasions by the Japanese during the Imjin War, and the Manchu during the Manchu invasions (1627–1636). Modern scholars have therefore claimed that, seeking refuge from political upheaval, the mid-Chosŏn literati increasingly engaged with Daoist themes and idyllic works of art that brought emotional comfort (Jungmann 2014, pp. 71–72). Although this may be true in some circumstances, the fact that prominent mid-Chosŏn scholars such as Yi I and Sŏng Che-wŏn, as well as late Chosŏn scholars such as Yi Ch'ŏn-sang, *documented* their spiritual experiences suggests that their journeys were not simply an act of withdrawal from society, but also a way of experiencing and recording spiritual progress (Ahn 2018, p. 120; Yang 2012, pp. 213–14).

Most importantly, late Chosŏn intellectuals employed the terms *ch'ŏnsŏn* 天仙 (divine transcendent beings) and *tongch'ŏn* 洞天 (cave-heavens, or heavenly abodes) to define realms on the peninsula as immortal lands. Cultured individuals accepted these terms quite literally, referring to individuals who possessed magical powers and resided in mythical lands as *sŏn* 仙, and referring to the residence of these individuals as *tongch'ŏn* 洞天, an auspicious place on earth serving as the entrance and passageway into the real *tongch'ŏn* (Verellen 1995, pp. 271–73). Literati like Sin Ik-sŏng 申翊聖 (1588–1644), who was King

Sŏnjo's 宣祖 (1552–1608) son-in-law, and poet Hwang Hyŏn 黃玹 (1855–1910) used the terms *tongch'ŏn* as well as *pokchi* 福地 (i.e., blissful realms where immortals reside) to describe scenic places they knew about from hearsay or had encountered during travel.[26] Erudite scholars imagined that by entering an earthly *tongch'ŏn* or *pokchi*, they would be able to reach the real immortal lands where they would encounter and even become transcendent beings. As a result, valleys in Mt. Kŭmgang such as Ten Thousand Falls Ravine (Manp'oktong 萬瀑洞) or Jade Stream Ravine (Ongnyudong 玉流洞) have the term *tong* 洞 incorporated into their names, which I believe are related to the Daoist concept of *tongch'ŏn*. In the geographical context of Mt. Kŭmgang, *tong* take the form of a ravine rather than a cave. Literate travelers applied the concept of *tong* to these valleys primarily to transform the terrain into an otherworldly place that would allow them to engage in self-cultivation activities.

In their travel accounts, literati "documented" more proof of Mt. Kŭmgang's otherworldly nature by the discovery and confirmation of otherworldly signs such as the nest of a crane, an important Daoist motif and symbol of longevity. There was also a cliff named Crane Nest Terrace (Haksodae 鶴巢臺) near Wŏnt'ong Hermitage (Wŏnt'ongam 圓通庵) in Inner Kŭmgang (Naegŭmgang 內金剛) where elite travelers would search for a genuine crane's nest to confirm Mt. Kŭmgang as an otherworldly place.[27] Many sixteenth and seventeenth century elite travelers who visited other mountains in southern Korea tell similar stories about crane nests, adding further evidence to the notion that seeing traces of otherworldly space was an essential component of elite travel.[28]

A chessboard cut into the mountain's rocks for visiting transcendent beings was a human-made landscape feature that enhanced the otherworldliness of a scenic location. Unknown travelers or mountain residents had such a chessboard carved into the flat riverbed rock at Mt. Kŭmgang's Ten Thousand Falls Ravine (Figure 11). The anonymous carver titled it Three Mountains' Chessboard (Samsan'guk 三山局). According to mythology, transcendent beings from three sacred mountains (a reference to the three mythical islands that were the homes of immortals, and were also believed to exist on the Korean peninsula, see below's discussion of the government garden in Wŏnju) would come to Ten Thousand Falls Ravine to play chess periodically. In contrast to the 9 × 10 gameboard of traditional Korean chess, this stone-carved board appears to be a 20 × 20 gameboard, which supports the idea that it was not made for human use but for otherworldly beings. More than twenty of such chess boards were cut into the rocks of mountains in Korea during the Chosŏn, including Mt. Sobaek (Sobaeksan 小白山) in Ch'ungch'ŏng Province (Ch'ungch'ŏngdo 忠淸道) and Mt. Tŏk (Tŏksan 德山) in Chŏlla Province (Chŏllado 全羅道), and are mentioned in travelers' records, indicating that the chessboard was a common theme that reinforced travelers' interpretation of those mountains as otherworldly.

The above-discussed examples of source material—either published in scholars' literary writings and/or carved into the landscape—vividly show that the late Chosŏn elite created mystical lands in Korea where they could experience the idealized life of a transcendent being, building on Daoist cosmological ideas. This late Chosŏn Korean elite practice of building immortal lands corresponds to practices in late Imperial China where high officials and emperors built luxurious gardens to practice Daoist rituals and self-cultivation.[29] Late Chosŏn Koreans were most familiar with these late imperial Chinese interpretations of Daoist ideas and figures, which entered Korea through the import of various encyclopedia, novels, poetry collections, travel records, and painting manuals, reflecting Ming's broader intellectual impact on Chosŏn (Pak 2011; Kim 2019). A marvelous example that shows how entrenched the idea of creating a Daoist land of immortals in Korea was to the elite is discussed in the final section of this article, which focuses on a rare example of premodern Korean garden culture. The garden was established by Kangwŏn Province governors at the back of their government residence, the Kangwŏn Provincial Office (Kangwŏn kamyŏng 江原監營) in Wŏnju 原州, some 90 miles south of Mt. Kŭmgang. Kangwŏn gov-

ernors created, renovated, and expanded this garden over the span of more than 200 years, transforming it into a perceived land of immortals.

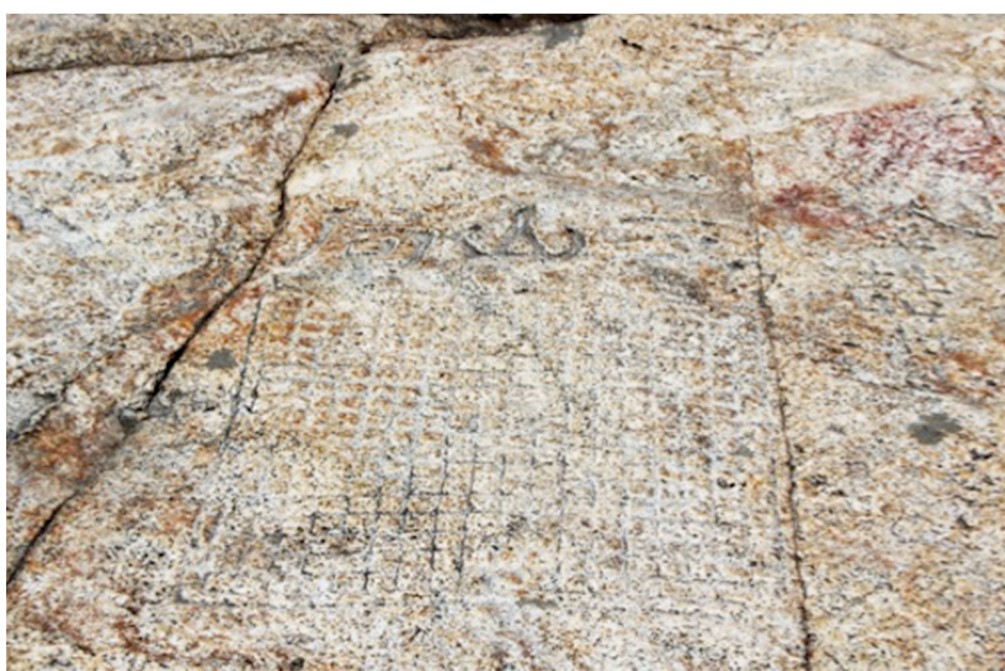

**Figure 11.** *Rock carving of a chessboard titled Samsan'guk*, Chosŏn period. Ten Thousand Falls Ravine, Inner Kŭmgang, Mt. Kŭmgang, DPRK. Photograph by author.

## 5. A Garden's Design Recalling Korea's Immortal Land of Penglai

During the late Chosŏn, the governor of Kangwŏn Province went on one-month provincial tours twice a year, in spring and autumn, during which he was able to visit Mt. Kŭmgang and other mountains in the province.[30] For the rest of the year he was confined to the city of Wŏnju, where the provincial government headquarters was located. Therefore, modern scholars have suggested that the governors created a back garden (*huwŏn* 後園) to temporarily escape administrative duties and entertain their guests (Yi 2016). While this may be true, my research reveals that Kangwŏn governors placed high value on the names of buildings when they designed and expanded their office garden, which was excavated in 2013 and reconstructed in 2017.

Kangwŏn governors built and expanded the Wŏnju garden with strong Daoist underpinnings. The garden's history begins in 1684, when then governor Sin Wan 申琓 (1646–1797) built a six bays (*k'an* 間) wide pavilion north of a 1400 square meter (roughly 15,000 square feet) large pond, and named it Penglai Pavilion (Pongnaegak 蓬萊閣). Sin wrote on Penglai Pavilion's beam that Kangwŏn was the home of transcendent beings since their abode, Mt. Penglai (=Mt. Kŭmgang), was located in the province.[31] Additional structures were built by governors in the eighteenth and nineteenth centuries, their names revealing the governors' Daoist reading of their garden's physical setting. For example, in 1746, governor Kim Sang-sŏng 金尙星 (1703–1755) added the Pavilion for Calling Transcendent Beings (Hwansŏngjŏng 喚仙亭) as a welcome place for immortals, and in 1771, then governor Sŏ Myŏng-sŏn 徐命善 (1728–1791) added Gathering Medicinal Herbs Dock (Ch'aeyago 採藥塢). The dock's name originates from the legend of the three islands where immortals gathered herbs of eternal youth. These new buildings were designed to heighten the sense of immersion in the land of immortals.

The *Kwandong Gazetteer* (*Kwandong chi* 關東誌), published in 1820, features a stunning illustration of the Kangwŏn Provincial Office, with the garden's layout prominently displayed in the upper left corner (Figure 12). Two of the three fabled islands in the Bohai 勃海 sea are represented by two small islands inside a large, square-shaped pond:

Yingzhou Pavilion (Yǒngjusa 瀛洲榭) and Penglai Pavilion (Pongnaegak 蓬萊閣). In the late nineteenth century, the garden also included Fangzhang Terrace (Pangjangdae 方丈臺), which refers to the third of the three islands, and the Pavilion for Terrapin Catch (Choojǒng 釣鰲亭), a term originating from the legend of the three mountains where Daoist immortals spent blissful hours catching terrapins at a pavilion.

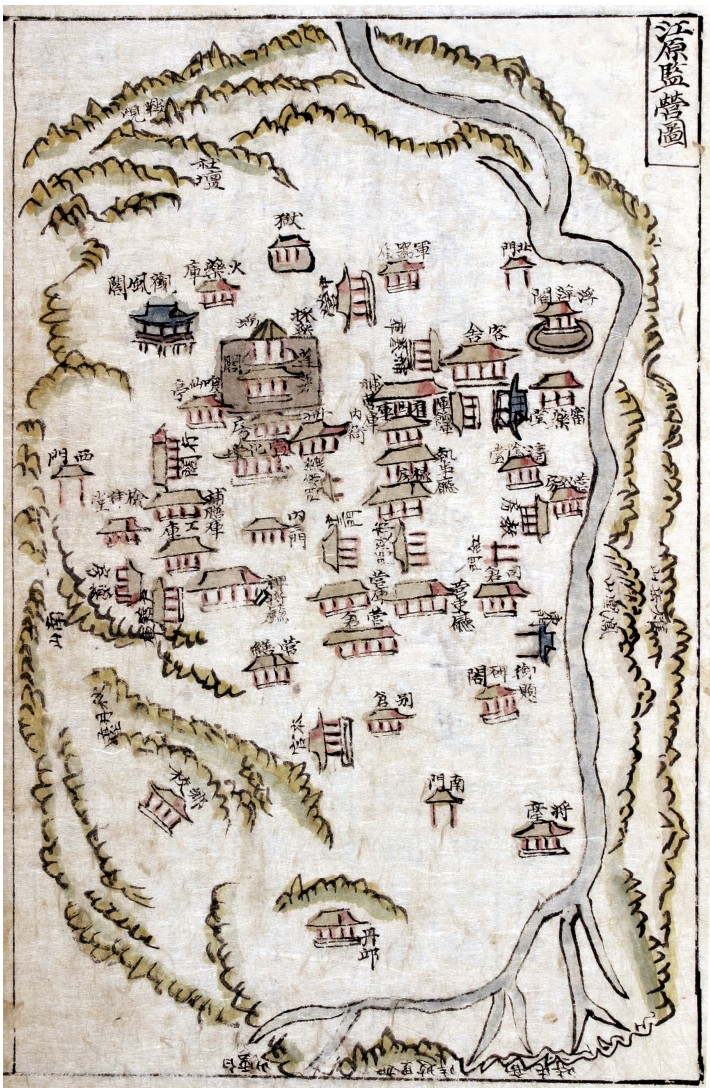

**Figure 12.** *Map of Kangwǒn Provincial Office*, *Kwandong Gazetteer*, Chosǒn period, 1820, Wǒnju City History Museum. Reproduced with permission from the Wǒnju City History Museum.

The grand scale of the Wǒnju garden with its large lotus pond and several islands (an extended version of the typical Chosǒn structure of a round-shaped island in a rectangular pond, *pangji wǒndo* 方池圓島) provided a distinctly different visual experience than what upper class individuals would have generally perceived in an aristocratic residential garden setting. An aristocratic residence could have had a combination of gardens such as a kitchen garden, a fruit orchard, and a rock garden with a small lotus pond that was typically located near the men's quarters, the *sarangch'e* 舍廊梗, as exemplified by gardens at Sǒngyojang 船橋莊 in Kangnǔng, Kangwǒn Province, and Sǒsǒkchi 瑞石池 in Yǒngyang, Kyǒngsang Province ([Min 1991](#), pp. 242–75). In contrast, governors had the resources to fund labor-intense construction of very large ponds with several islands. Consequently, large gardens such as the one in Wǒnju were primarily funded by the state. A comparable example is Kwanghallu 廣寒樓 garden in Namwǒn, Chǒlla Province. In 1444, Chǒlla gov-

ernor Chŏng In-ji 鄭麟趾 (1396–1478) had three large islands constructed inside Kwang-hallu's pond. The largest island, referred to as Pongnae, measured 158 square meters (roughly 1700 square feet). The structure was continuously renovated and repaired by late Chosŏn magistrates of Namwŏn (Min 1991, pp. 233–34; Pak 2010, p. 311–17).[32]

The idea that the three legendary islands were also located on the Korean peninsula likely inspired the creation of the Wŏnju and Kwanghallu garden. As discussed above, Mt. Kŭmgang was believed to represent Mt. Penglai. Fangzhang was represented by Mt. Chiri in the southwest, while Yingzhou was represented by Mt. Halla on Cheju Island. It was also believed that Mt. Chiri was the residence of Taiyi 太乙 (the primordial unity of *yin* and *yang*) and therefore the place where transcendent beings convened (Yi 2019, p. 131). The architectural structures in these gardens thus demonstrate that governors and magistrates not only desired to create a garden for relaxation and entertainment, but they also wanted to build an environment that would facilitate spiritual rejuvenation and cultivation. In order to do so, the government-officials followed an established literary pattern of trying to localize an ancient Chinese myth in the Korean landscape by naming things and places accordingly, and making the names present at the garden in the form of name-boards (cf. Clunas 1996, pp. 144–48). In the case of the Wŏnju garden, numerous late Chosŏn scholars not only applied this rhetoric to the naming of the garden's pavilions but also used terms such as governor of Pongnae (Pongnae kwanch'alsa 蓬萊觀察使) or Earl of Pongnae (Pongnaebaek 蓬萊伯) as an alternative title for the governor of Kangwŏn. Yi Ch'i-jung 李致中 (1726–1802), Kangwŏn governor in 1786, even called himself owner of Pongnae (Pongnae chuin 蓬萊主人) when he composed a poem while riding in a boat on the garden's pond (Yi 2016, pp. 27–28, fn51).

The Kangwŏn governors' literary appropriation of the Wŏnju garden, and by extension of Pongnae/Penglai, reveals a strategy to exploit the notion of an otherworldly land as a reference to their spiritual aspirations.[33] Calling himself a recumbent transcendent being (*wasŏn* 臥仙), Sŏ Myŏng-sŏn for example wrote that "even if not getting wings and rising as a transcendent being, when I am at this garden, I get the feeling of being in the world of immortals." And Im Han-ho 林漢浩 (1752–1827), who was appointed governor of Kangwŏn in 1805, composed the following poem in which he alludes to the government garden as a microcosmos that contains Mt. Pongnae:

> Despite dwelling in the world, not being defiled by it
>
> A beautifully adorned residence of transcendent beings, [a waterway] specially opened
>
> Floating all day long, the wind carrying the boat
>
> The sky reflecting in the rippling water, encircling the platform
>
> By cutting sacred mountains, the three islands are complete
>
> From the azure-blue sea, pouring a glass of water
>
> Through high costs and people's meritorious deeds, [heaven] responding with harmony
>
> In a small universe, there is a small Pongnae (Yi 2016, p. 25)

In this poem, Im Han-ho comments on the man-made waterscape inside the garden. Some of the lines suggest how water transformed the garden into the islands of immortals. The phrase "specially opened" implies that an inlet was constructed to transport water into the garden's pond. In fact, archaeological excavations revealed such inlet (see Figure 13). The line "from the azure-blue sea, pouring a glass of water" praises the pond as if it were from the azure-blue sea (where immortals live). The following line, "through high costs ...," emphasizes that the creation of this waterscape was costly and required much man-power, yet the water allows the garden to be transformed by Heaven. This poem is relevant for our discussion because it demonstrates that Kangwŏn governors valued not only mountains but also waterways as key components of their (artificial) land of immortals.

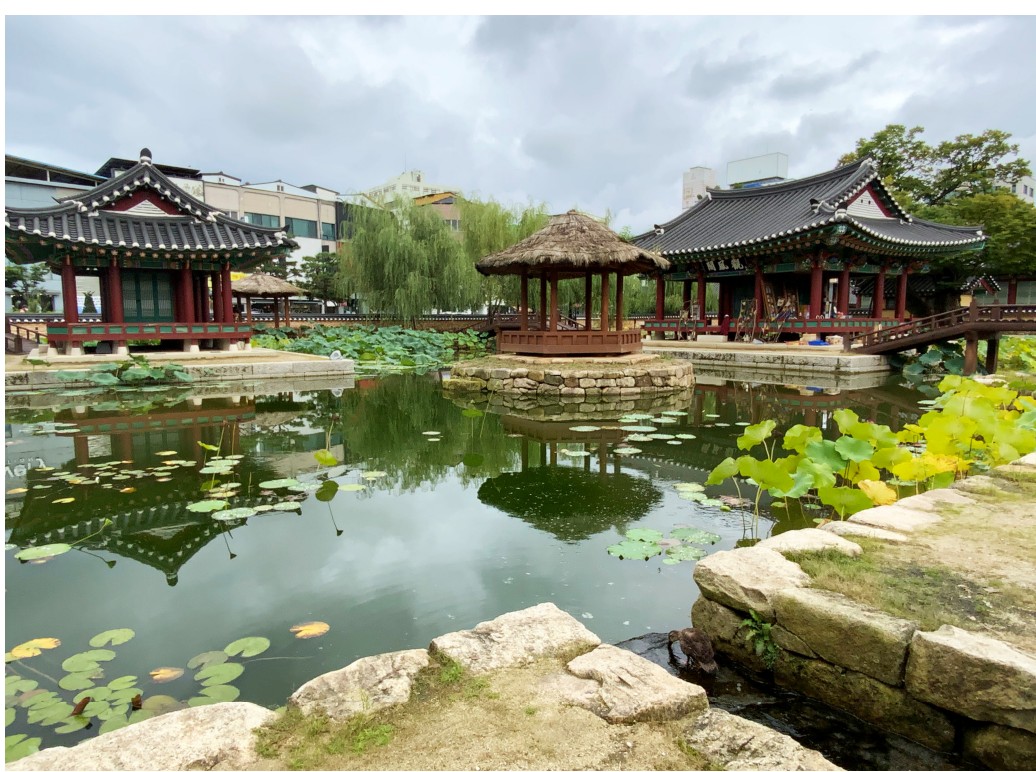

**Figure 13.** Back Garden of Kangwŏn Provincial Office. Late Chosŏn Dynasty, modern reconstruction based on archaeological excavations. Buildings from left to right: Pongnae/Penglai Pavilion, Pavilion for Terrapin Catch (thatched; behind Pongnae Pavilion), Gathering Medicinal Herbs Dock (thatched round structure), and Yŏngju/Yingzhou Pavilion. Inlet on lower right of picture. Wŏnju, Kangwŏn Province. Photograph by author.

On the other hand, governor Kim Sang-sŏng 金尙星 (1703–1755) strongly emphasized the Wŏnju garden's connection to Korea's Mt. Penglai, i.e., Mt. Kŭmgang. Upon his arrival in Wŏnju, he began making several enhancements to the garden, starting with the restoration of Pongnae Pavilion. To reinforce the idea of entering an earthly paradise while sitting in the newly restored pavilion and enjoying the view of the pond, Kim ordered a rubbing of Yang Sa-ŏn's calligraphy (carved at Mt. Kŭmgang's Ten Thousand Falls Ravine) and had it hung on Pongnae Pavilion's wall, and he also commissioned a painting of Mt. Kŭmgang scenery. Together with his guests, Kim strolled through the garden while singing the Kwandong Melody (Kwandonggok 關東曲) and fantasizing about roaming in the world of immortals.[34] Kim essentially incorporated historical traces and visual imagery of Mt. Kŭmgang into the garden's space to validate the garden as the epitome of an otherworldly land on Korean soil.

## 6. Concluding Reflections

This article pioneers inter-disciplinary research that hopefully will inspire further investigation into the ways in which late Chosŏn Koreans encountered and practiced various aspects of Daoism. So far, art historians and literature scholars have explored Chosŏn period Daoist material in the context of withdrawal from society or intellectual entertainment. However, various examples from textual as well as visual and material culture demonstrate that Daoist themes pervaded late Chosŏn society, from Daoist ritual texts and icons at various religious shrines to the construction of artificial mountains for theater performances and the establishment of Daoism-inspired landscapes and government gardens, which show that Chosŏn literati's self-cultivation practices were inspired by actual as well as artificial landscapes. The research also shows that in Korea, Daoist icons were not part of a separate Daoist institution, but were rather absorbed into a variety of religious and

cultural contexts both private and official in nature. This article thus brings Chosŏn Korea into the conversation about the various roles of Daoism in East Asian cultures in which it has, unfortunately, all too often been absent.

**Funding:** The article processing charges related to the publication of this article were supported by The University of Kansas (KU) One University Open Access Author Fund, sponsored jointly by the KU Provost, KU Vice Chancellor for Research & Graduate Studies, and KUMC Vice Chancellor for Research and managed jointly by the Libraries at the Medical Center and KU – Lawrence, and The Kress Foundation Department of Art History of the University of Kansas (KU).

**Acknowledgments:** I would like to thank the anonymous reviewers of this article and previous versions of it for their valuable feedback. I would also like to thank the graduate students in my Korean Painting seminar in Fall 2022 for being ardent readers and respondents to this article. Special thanks go to Vincent Gossaert for reminding me of the most recent scholarship on Guan Yu and pointing me to terminology issues. Any remaining errors or oversights in this article are all mine.

**Conflicts of Interest:** The author declares no conflict of interest.

## Notes

1  Glomb (2016); C.-h. Kim (2015); Kim (2001, 2013); Seo (2014, 172ff). For a historiographical overview on Daoism see Robson (2015); for a brief history of Korean Daoism see Jung (2000).

2  For an excellent overview on Daoist visual culture in the imperial Chinese context see Huang (2014, p. 929).

3  Hŏ Mok, "Musul chuhaenggi" 戊戌舟行記, 15:6a; an entry in the *Sinjŭng Tongguk yŏji sŭngnam* 東國輿地勝覽, printed in 1530–31, also records that Xue Rengui 薛仁貴 was worshipped as the mountain spirit of Mt. Kamak, see Yi Haeng, "Kyŏnggi Chŏk-sŏnghyŏn" 京畿積城縣. In *Sinjŭng Tongguk yŏji sŭngnam* 新增東國輿地勝覽, 11:42b.

4  Unless otherwise noted, the word provided in parentheses is the romanized Korean term followed by original Sinitic characters; (S.) signifies a Sanskrit, (C.) signifies a Chinese term. All translations in the text are mine unless noted otherwise.

5  *Dafangguang fo huayan jing* 大方廣佛華嚴經, T09n0278p0395, see https://cbetaonline.dila.edu.tw/zh/T0278_001 (last accessed on 20 September 2022); Kim (1997a, p. 213); McBride (2008, pp. 133–35) (list of deities based on the 60-volume version). From the Koryŏ dynasty (918–1392) onward, *sinjung* were worshipped as a separate group of deities, as indicated by the construction of a Sinjung Cloister (Sinjungwŏn 伸衆院) in the Kaegyŏng area in 925, and the performance of *sinjung* rituals at least 37 times during the thirteenth century to pray for the protection of the country against the Mongol invasions (1231–1270) (Yi Kyubo, *Tongguk Isanggukchip* 東國李相國, 41:1a-b; Kim 1994, p. 198). Until the sixteenth century, the royal court held Daoist rituals to heaven and stars for the avoidance of disaster and bestowal of good fortune for the state. At the Sogyŏksŏ 昭格署, officials worshiped the Jade Emperor (Okhwang sangje 玉皇上帝), Laozi (Noja 老子), and King Yama (Yŏmnawang 閻羅王), see *Myŏngjong sillok* 明宗實錄 5:70a (5/26/1547). The Sogyŏksŏ was not rebuilt after its destruction during the Imjin War, see Anonymous, *Tongguk yŏji pigo* [東國輿地備考], 1:51a. But the royal court continually held ceremonies at the Tangun Shrine (Tan'gunsa 檀君祠) in P'yŏngyang and elsewhere to worship Korean rulers of antiquity including Tan'gun 檀君, Kija 箕子, and King Tongmyŏng (Tongmyŏngwang 東明王), see *Sejong sillok chiriji* 世宗實錄地理志, P'yŏngyang 平壤 section, 154:2b (1454); *Sukchong sillok* 肅宗實錄 31:39b (7/4/1697); *Yŏngjo sillok* 英祖實錄 49:23b (5/23/1739).

6  Images of Skanda first appear in late Koryŏ woodblock-printed publications, but during the Chosŏn he was no longer only a protector of Mahayana Buddhist texts but played a major role in the development of *sinjung* banner paintings, see Kim (2021, pp. 67–80).

7  For the iconography of the mountain god and the kitchen god in *sinjung* paintings, see H.-j. Kim (2015, pp. 57, 61).

8  Ku Mirae, "Pulgyo sesi ŭirye ro pon sinjung sinang ŭi Han'guk chŏk suyong [The Korean acceptance of the faith in guardian deities as seen in Buddhist seasonal rituals]," p. 152.

9  One aspect that scholars in the field of Chinese religions seem to emphasize in their work is the ways in which different social groups contested over and/or regulated the worship of specific deities on the local level (Goossaert 2014; Naquin 2000; Szonyi 2007). Scholarship on Korean religions yet needs to explore such issues more broadly.

10  *Sukchong sillok* 肅宗實錄 38:61b (6/18/1703)

11  *Sukchong sillok* 肅宗實錄 23:9b (2/26/1691) and 23:10a (2/27/1691); cf. Kim (2018, p. 293).

12  The late Chosŏn understanding of a loyal and righteous Guan Yu seems to mirror late imperial Chinese interpretations of this deity, see Ter Haar (2000, p. 203).

13  Maurice Courant, *Bibliographie coréenne: tableau littéraire de la Corée, contenant la nomenclature des ouvrages publiés dans ce pays jusqu'en 1890*, pp. 151–95; Kim (2022, pp. 296–97; 2020, p. 10; 2014, p. 164).

14  Wan (2015, p. 47); Schipper, "Shenxiao Fa and Related Thunder Rites" in Schipper and Verellen (2004, vol. 2, p. 1092); cf. Huang (2012, pp. 250–51).

[15] Yun (2014, p. 274); Wan (2015, figs. 11/12). It remains to be explored which Chinese versions served as models for Chosŏn period versions of this scripture.

[16] For the popular use of the Precious Jade Pivot scripture and the Jade Pivot Scripture in late Chosŏn see *Yŏngjo sillok* 102:20b (1763/9/28) and Jung (2000, p. 814ff). It remains to be explored how the function of this scripture changed in royal rituals over the course of the 500-year long history of the Chosŏn dynasty; some Daoist rituals were apparently re-introduced into court culture during the late eighteenth and nineteenth centuries, see endnote below. For the 1906 abolition of rituals at the Jade Pivot Shrine see *Sunjong sillok* 2:21a (7/23/1906). About shamanic rituals during which the *Precious Jade Pivot Scripture* was recited see Nam (2016, p. 56).

[17] Chosŏn period scholars who studied *naedan* include Kim Si-sŭp 金時習 (1435–1493), Chŏng Hŭi-ryang 鄭翬良 (1706–1762), and Sŏ Myŏng-ung 徐命膺 (1716–1787), see Chŏng (2006, pp. 36–37); for Chosŏn period writings about *naedan* practice see Kim (2012). Daoist cultivation practices of mind and body were the basis for several medicine manuals such as the *Ŭibang yuch'wi* 醫方類聚, completed in 1445, and the *Tongŭi pogam* 東醫寶鑑, which was written by royal physician Hŏ Chun 許浚 (1539–1615), and was published by the royal court in 1610.

[18] The *Cantong qi* 參同契 (Seal of the Unity of the Three) is one of the oldest and most important Chinese texts on inner alchemy.

[19] Kang Hŏn-gyu, *Nongnyo chip*, 5:7a: "Yu Kŭmgangsan rok." In this article, I use the terms "immortals" and "transcendent beings" interchangeably when referring to *sŏn* 仙 (C. *xian*). In the field of East Asian art history, 仙 is generally translated as "immortals" but in the recent scholarship on Chinese Daoism the preferred translation is "transcendent" or "transcendent being" to emphasize the metaphysical aspects of Daoism which aimed toward transcendence of the individual. I agree that "transcendent" is a more accurate translation of 仙 but particularly for standard phrases such as the "three mythical islands where immortals live" I decided to use the more established term "immortal."

[20] The term *sandae* discussed here is not to be confused with the term *sandae nori* which refers to a particular type of mask dance from the Seoul region, see Sa (2002, p. 376 ff).

[21] Another reason could have been corruption issues, see Ahn (2010, p. 257).

[22] For travel writers, the claiming of unknown places and the rectification of names were core functions of the social elite, see Strassberg (1994, pp. 6, 21); Harrist (2008, p. 18).

[23] Sŏng Che-wŏn, "Yu Kŭmgangsan ki" 遊金剛山記, p. 329 (稿中:10b). For a discussion of the ways in which local scholars in Yŏngnam 嶺南, the region of present-day Kyŏngsang Province, conceptualized space primarily using Confucian ideology, see Chŏng (2012). For research on the connection between poetry composition and so-called pavilion culture, see Pak (2006).

[24] Yi Ch'ŏn-sang, "Kwandongnok" 關東錄, p. 465.

[25] Yi I, *Yulgok Sŏnsaeng chŏnsŏ* 栗谷先生全書, p. 56 (拾遺 1:28a).

[26] Sin Ik-sŏng, "Yu Kŭmgang sogi" 遊金剛小記, 7:33a; Hwang Hyŏn, "Obong sin'gŏ Sangnyangmun" 五峯新居上梁文, 2:16a.

[27] Yi Man-bu, "Kŭmgangsan ki" 金剛山記, 3:11a; An Kyŏng-jŏm, "Yu Kŭmgangnok" 遊金剛錄, p. 145.

[28] Mt. Kŭmgang's crane nest is mentioned for example by Hong Kyŏng-mo, "Haeakki" 海嶽記, p. 1061; for travelers associating the crane story with scenic locations at other mountains, see Chuwangsannok 周王山錄 by Chang Hyŏn-gwang 張顯光 (1554–1637); Yu Naeyŏngsannok 遊內迎山錄 by Hwang Yŏ-il 黃汝一 (1556–?); and Wŏlmaksansugi 月幕山水記 by No Kyŏng-im 盧景任 (1569–1620).

[29] Within the Sinitic cultural realm, immortal lands were traditionally considered the perfect environment to achieve union with the Dao by refining alchemical drugs, see Wan (2009, p. 68) (footnote 12); for cave-heavens see also Stein (1990, 73ff).

[30] Early Chosŏn period governors were on inspection rounds (*sullyŏk* 巡歷) during their entire time in office, but in the late seventeenth and eighteenth century, this system changed and the governor stayed mostly at his government office, see Kungnip Munhwajae Yŏn'guso (2016, p. 97).

[31] Taehan Min'guk Mun'gyobu Kuksa P'yŏnch'an Wiwŏnhoe, *Yŏji tosŏ* 興地圖書, p. 493.

[32] Further research will provide more details about the Daoist implications of Kwanghallu. A slightly different example of a magistrate's garden with two rectangular islands in a rectangular pond (*pangji ssangbangdo* 方池雙方島) was depicted by Chŏng Sŏn 鄭歉 (1676–1759) in his painting of Ssangdo Pavilion (Ssangdojŏng to 雙島亭圖), which is currently in a private collection.

[33] My interpretation of the material differs from an argument expressed by Yi Sang-gyun, according to which governors maintained the Wŏnju garden purely for entertainment and relaxation, see Yi (2016, pp. 26–28).

[34] Anonymous, "Pongnaegak" 蓬萊閣, 4: 26–27.

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
