# Peer review of "Warrior Gods and Otherworldly Lands: Daoist Icons and Practices in Late Chosŏn Korea"

_religions, doi:10.3390/rel13111105_

Round 1
Reviewer 1 Report
I think this article is fine, just as long as the sentence in lines 38-39 is revised. It is unclear, perhaps as the wrong word ("literate" instead of perhaps "literature") and seems to be missing a verb or some other word. As currently written it is not correct English.
Author Response
Thanks so much for reviewing my manuscript. In lines 38-39 I was using the term "literate" as opposed to "illiterate," but I understand that it could be confusing, so I changed it to "literate people."
Reviewer 2 Report
“Warrior Gods and Otherworldly Lands: Daoist Icons and Practices in Late Chosŏn Korea” is a well-constructed essay on a topic little written about and not well known in the field of Korean religions. This essay does an admirable job of articulating the role of Daoist gods on the Korean peninsula.
There are only a few places where the author needs more or better evidence, or perhaps more contextualization. In line 182, the author writes that “Kwanje had evolved into the protector deity of the Chosŏn royal house.” Does the author mean “the” as in their being only one protector deity—or was Kwanje “a” protector of the Chosŏn royal house? This is a significant claim. What is the evidence supporting this assertion? The rise of icons of Kwanje is important—but what is the evidence that this deity was claimed exclusively by the royal family of Chosŏn?
The author should be consistent using English translations of the names of Daoist deities. Most of the time the author uses English translations, but for some reason uses Puhua tianzun instead Supreme God of Thunder (pp. 7-8), for the Daoist form of Samantabhadra. In lines 251 and 252, the author uses Puhua tianzun and then Highest Prince of Jade Purity. Both should be in English, unless the author has some compelling reason for using Puhwa tianzun. If so, the author needs to explain why this is necessary.
The author uses the McCune-Reischauer Romanization system in this essay. Typically, in the best scholarship, scholars translate translatable terms such as -sa 寺 as “monastery,” -san 山as “mountain,” -dae 臺 as “terrace,” and -am 庵 as “hermitage” instead leaving these common terms untranslated. The author usually does this for Chinese place names, such as on p. 15. Why doesn’t the author do this for Korean place and geographic names? The author should consider doing this throughout the paper to facilitate understanding—and more importantly to be consistent.
This reviewer’s reading of scholarship on Chinese Daoism is that in recent decades, and certainly during the 21st century, scholars prefer to use “transcendent” or “transcendent being” as a translation of xian 仙 (Kor. sŏn). However, this author does not seem particularly familiar with this trend in Western scholarship on Daoism because he/she uses “heavenly immortals” or “celestial being” for ch’ŏnsin 天仙 (line 398–399)
Line 75: Xue Ren-gui => Xue Rengui
Line 75: Kam’aksan => Kamaksan. Is there a reason for not using Mt. Kamak?
Line 152: Hideyoshi invasions; in more recent scholarship, scholars use “Imjin War.” It may be useful to have both.
Line 168: Romance of the Three Kingdoms should be italicized.
Line 182: the protector deity of the Chosŏn royal house => a protector deity of the Chosŏn royal house.
Line 201: Romance of the Three Kingdoms should be italicized.
Line 218: Teaching of Heavenly Principles => Teaching of the Heavenly Way (or sometimes Ch’ŏndoism)
Line 226: Yushu baojing should be italicized.
Line 227: Okchu pogyŏng should be italicized.
Line 227: Jiutian yingyuan leisheng puhua tianzun yushu baojing should be italicized.
Line 232–233: Use English translation Supreme God of Thunder and put Chinese name in parentheses (Puhua tianzun).
Line 251: Use Supreme God of Thunder instead of Puhua tianzun.
Line 267: Ch’amdonggye chuhae should be italicized.
Line 272: Toga chichi tokcho kyŏng should be italicized.
Line 287: Is there a reason or not writing “on Mt. Kŭmgang”?
Line 324–325: Fengshitu could be translated as Illustrations of an Imperial Commissioner (Fengshitu).
Line 363: Orthodox scholar Yi Ch’ŏn-sang => The orthodox Confucian scholar Yi Ch’ŏn-sang.
Line 366: Kŭmgangsan => Mt. Kŭmgang
Line 380: Kŭmgangsan => Mt. Kŭmgang
Line 398: ch’ŏnsŏn should be translated as “divine transcendents” or “divine transcendent beings”
Line 420: Crane Nest Platform should be “Crane Nest Terrace”
Line 431: Kŭmgangsan => Mt. Kŭmgang
Line 426 and 431: Samsanguk => Samsan’guk
Line 440: note xliii is not found in note section
Line 454 and 459–460: Kŭmgangsan => Mt. Kŭmgang
Line 468: Pongnae Pavilion => Penglai Pavilion; Penglai is a term more well known to Anglophone audiences than Pongnae; although both terms refer to the same mythical island.
Line 470: Pongnaesan (=Kŭmgangsan) => Mt. Pongnae (=Mt. Kŭmgang)
Line 482: Kwandongji => Kwandong chi and it should be italicized
Line 485–486: Yingzhou Terrace 瀛洲榭 (K. Yŏngjusa): a sa 榭 (Ch. xie) is not a terrace; it is a house or pavilion on a terrace. So, it would be better to call it a “pavilion.” Are these the Chinese names for these small islands, or are they shared by both China and Korea? The author needs to be consistent.
Line 494: Kŭmgangsan => Mt. Kŭmgang
Line 495 and 497: Chirisan => Mt. Chiri
Line 496: Hallasan => Mt. Halla
Line 506: Pongnaebaek 蓬萊伯 (Duke of Penglai/Pongnae) should be (Earl of Penglai/Pongnae)
公 = Duke; 侯 = Marquis; 伯 = Earl; 子 = Vicount; 男 = Baron
Hebo 河伯 (Kor. Habaek), the God of the Yellow River, is the “Earl of the River”
Line 542 and 547: Kŭmgangsan => Mt. Kŭmgang
Line 544 and 548: Pongnaegak => Penglai Pavilion or Pongnae Pavilion
Line 550: Kwandonggok should be translated as “Kwandong Tune” or “Kwandong Melody”
Line 554: Concluding remarks => Concluding reflections
“Remarks” are given when someone speaks; reflections works better in this case
Line 581: Kwandongŭpchi => Kwandongŭp chi
Line 587: Dafangguan fo huayan jing should be italicized
Line 589: Kiŏn pyŏljip => Kiŏn pyŏlchip (this is correct McCune-Reischauer because Sino-Korean word)
Line 604: Nongnyojip => Nongnyŏ chip (and italicized)
Line 610: “Yu Kŭmgangsan’gi” => “Yu Kŭmgangsan ki”
Line 616: Sinjung Tongguk Yŏji Sŭngnam => Sinjŭng Tongguk yŏji sŭngnam
Line 621: “Tongyurok” => “Tongyu rok”
Line 711: McBride, Richard D. => McBride, II, Richard D.
Line 712: Hawaii => Hawai‘i (use open single quote as okina)
p. 24, n. xxiv: Tongŭi pogam should be italicized.
1610. - Chosŏn period => delete hyphen
p. 25, n. xliii is missing????
Author Response
Thank you very much for your comments. I truly appreciate that you took the extra time and effort to thoroughly review my manuscript. Please see attached file for my responses to each of your comments. I edited the entire manuscript following your suggestions.
